# Neuropeptide S receptor gene Asn107 polymorphism in obese male individuals in Pakistan

Aftab Ahmad[1], Fahad Almsned[2,3], Pasha Ghazal[1], Malik Waqar Ahmed[1], M. Saleet Jafri[2,4], Habib Bokhari[1,4]*

1 Department of Biosciences, COMSATS University Islamabad, Chak Shahzad, Islamabad, Pakistan,
2 School of Systems Biology and Krasnow Institute for Advanced Study, George Mason University, Fairfax, Virginia, United States of America, 3 King Fahad Specialist Hospital– Dammam, Dammam, Saudi Arabia, 4 Center for Biomedical Engineering and Technology, University of Maryland School of Medicine, Baltimore, Maryland, United States of America

* habib@comsats.edu.pk

**Data Availability Statement:** All relevant data are within the manuscript and its Supporting information files.

## Abstract

Neuropeptide S (NPS) is a naturally occurring appetite stimulant, associated with anxiety, stress, and excitement regulation. Neuropeptide S serves as a hypothalamic energy regulator that enhances food intake with a reduced level of satiety. NPS activates fat angiogenesis and the proliferation of new adipocytes in obesity. NPS has an established role in energy regulation by many pre-clinical investigations; however we have limited data available to support this notion in humans. We found significant association of Neuropeptide S receptor (NPSR1) Asn107Ile (rs324981, A>T) polymorphism with obese male participants. The current investigation carried out genotype screening of NPSR1 allele to assess the spectrum of the Asn107Ile polymorphism in obese and healthy Pakistani individuals. We revealed a significant (p = 0.04) difference between AA vs TT + AT genotype distribution of NPSR1 (SNP rs324981,) between obese and healthy individuals ($p$ = 0.04). In this genotype analysis of (SNP rs324981) of the NPSR1 gene, T allele was marked as risk allele with higher frequency in the obese (38%) compared to its frequency in the controls (25%). Single Nucleotide Polymorphism (SNP, rs324981) Asn107Ile of NPSR1gene, that switches an amino acid from Asn to Ile, has been found associated with increased susceptibility to obesity in Pakistani individuals. Furthermore, molecular simulation studies predicted a lower binding affinity of NPSR1 Asn107Ile variant to NPS than the wild-type consistent with the genotype studies. These molecular simulation studies predict a possible molecular mechanism of this interaction by defining the key amino acid residues. However, a significantly ($p$<0.0001) lower concentration of NPS was recorded independent of genotype frequencies in obese subjects compared to healthy controls. We believe that large scale polymorphism data of population for important gene players including NPSR1 will be more useful to understand obesity and its associated risk factors.

**Funding:** The study was supported by the Higher Education Commision, Pakistan (Grant # IRSIP36-BMS71) received by Aftab Ahmad. The funders had no role in study design, data collection and analysis, decision to publish, or preparation of the manuscript.

**Competing interests:** The authors declare no conflict of interest.

## Introduction

Obesity, defined as an unnecessary accumulation of body fat that puts an individual's health to risk, is now invading the world populations as an epidemic disease that counts for 2.8 million deaths every year [1]. Traditionally obesity thought to be associated with the developed part of the world; however, now increased prevalence has also been observed in developing and low-income countries. The prevalence of obesity has increased almost three-fold between 1975 to 2016 [1]. Obesity is measured by body mass index (BMI). An individual with a BMI $\geq$ 30 is considered obese and at higher risk concerning their susceptibility to diseases and other associated risk factors. Furthermore, obesity is a multifactorial disease. Therefore, it is important to identify all possible risk factors including genetic factors, and in particular, Neuropeptide S (NPS), which is associated with energy regulation and homeostasis in humans [2].

NPS binds with the Neuropeptide S receptor (NPSR1) and modulates various cellular functions through its signaling cascade. The NPSR1 gene located on chromosome 7p14.3 encodes the G protein-coupled receptors (GPCRs) vasopressin/oxytocin receptor subfamily. Furthermore, the NPS-NPSR1 interaction regulates the downstream Mitogen-Activated Protein Kinase (MAPK) system and accelerates fatty acid oxidation to protect animals from dietary obesity. NPSR1 has different physiological roles like modifying eating habits, release of adrenocorticotropic hormone (ACTH), suppressing anxiety, fear, panic, and sleep regulation [3, 4].

Polymorphisms in the NPSR1 gene have been linked with panic disorders, asthma, inflammatory bowel disease (IBD), and rheumatoid arthritis [5–7]. A change of an amino (Asn107Ile) is enforced by single nucleotide polymorphism (SNP) rs324981 A>T. T-allele (Asn107Ile) of the NPSR1 gene has a ten-fold higher risk for associated disorders. Nervousness and anxiety-related disorders are also strongly associated with NPSR1 T-allele [8–10].

NPSR1 gene expression is also dependent on external factors, e.g. individuals with the TT genotype were found to be highly sensitive to the development of depression and anxiety due to their malnourished status during their early phase of life [11]. Moreover, the external environment variably influences the association of NPSR1 Asn107Ile between populations [12]. Earlier investigations have revealed that the NPS-NPSR1 system effects the bioactivity of the NPS peptide due to A/T single nucleotide polymorphism [13, 14]. Moreover, previous studies have not investigated the NPSR1 variants' interaction with their functional efficiency. The current investigation explores the possible association of NPSR1 Asn107Ile variant with NPS serum levels and the NPS-NPSR1 interaction through molecular simulation analysis in obesity. We used molecular simulation to explore how the Asn107Ile variant structure differs from wild-type to understand how the two forms are functionally different.

## Material and methods

### Study population

This investigation has been designed to explore the association of NPSR1 gene with obesity in adult (20 to 50 years) Pakistani population. A total of 116 adult subjects with their written consent were included in this investigation. This study was approved by the Departmental Ethical Review Board of University of COMSATS University Islamabad. All adult male participants visiting Social Security Hospital Islamabad from diverse urban settings and communities of Pakistan were enrolled between August, 2017-June, 2018. A homogenous ancestry background of the studied population was assured to limit the influence of population stratification on our investigation. Participants were categorized on the basis of their BMI scores in Kg/m$^2$ presenting Kg as weight and height in meters squared (m$^2$). The participants were weighed in Kg by

using standard calibrated weight machine and height was measured in (cm) by using stadi-ometer both height, and weight was measured without shoes.

All participants were divided into groups as per WHO guidelines, those with BMI equiva-lent to 30 Kg/m$^2$ or more were categorized as obese, and those with BMI equivalent to 29.9 Kg/m$^2$ or less were considered as a control group. Study exclusion criteria were 1) having non-Pakistani lineage; 2) having any infectious disease; 3) having some other medical complication.

Participant's blood samples were collected in serum separating vacutainers. After collection, each blood sample tube was left undisturbed at room temperature for 30 minutes and the resulting top layer of serum was transferred into clean tubes. These serum tubes were kept at -20˚C until further analysis. NPS was quantified by using a commercial Human NPS (Neuro-peptide S) ELISA kit Catalog No: E6618h (EIAab® USA).

## Lipid profile

Total cholesterol, LDL, HDL, triglycerides levels, and blood glucose level for all participants were determined by utilizing automated enzymatic analyzers (Cobas Integra 700; Hoffman-La Roche, Basel, Switzerland).

## Genotyping

Genotyping of the NPSR1 Asn107Ile polymorphism was performed utilizing the PCR-RFLP technique as reported previously [15]. DNA was isolated from blood samples using the TIA-GEN® DNA blood kit (Tiagen China) according to the manufacturer's protocol. Briefly, 500 μl of each homogenized whole blood was processed for DNA isolation from all samples. DNA quantity and quality was measured by Nanodrop spectrophotometer (Nanodrop Technologies, USA) and visualized by agarose gel electrophoresis. Isolated DNA was utilized for amplification of rs324981 in a 25 μl volume reaction using thermal cycler (Thermo Electron Corporation) and NPSR1 rs324981F Forward 5′–ACC CTG AAT GTA AGC ACT TGT 3′ and NPSR1 rs324981R Reverse 5—TGT CTC ATC ACA TTT GGA AGGT—3′ set of primer was used under the following conditions: 95˚C for 5 min, followed by 35 cycles of 94˚C for 30 sec (dena-turation), 55˚C for 30 sec (annealing), 72˚C for 1 min (extension), and 10 min at 72˚C for a final extension. Gel electrophoresis was executed on 2% agarose gel for 40 min at 90 Volts and 400 mA to visualize the resulting PCR amplicon of 169 bps under UV light utilizing the transil-luminator. Finally, the resulting amplicon was treated with restriction endonucleases (BsiEI) from a strain of thermophilic *Bacillus circulans* under the following set of conditions incubated at 37˚C for 3 hours and afterward at 80˚C for 20 minutes. Processed amplicons were run in 2.5% agarose gel and visualized in UV illuminator (Alpha Imager Mini Bucher Biotech, Basel).

## Sanger sequencing

Sanger sequencing technique was applied to confirm all PCR-RFLP results using Big Dye Terminator Cycle Sequencing Ready Reaction Kit and the ABI PRISM 3730 DNA analyzer (Applied Biosystems, USA).

## Enzyme-linked immunosorbent assay (ELISA)

Human NPS ELISA kit (EIAab® The Top Notch Elisa Kit Manufacturers USA) was used for the *in vitro* quantitative determination of human Neuropeptide S levels in serum of both obese and control groups. The samples were added to each microtiter plate for preparation of biotin-conjugated antibody specific for human NPS with Avidin conjugated to Horseradish

Peroxidase (HRP) and incubated for three hours, Wells with biotin-conjugated antibody and enzyme-conjugated Avidin exhibited change in color on the addition of a tetramethylbenzidine substrate solution. A change in color was measured at a wavelength of 450 nm ± 2 nm with a spectrophotometer after terminating enzyme-substrate reaction. The concentration of neuro-peptide S was determined by comparing the O.D. of each sample to the standard curve.

## Molecular simulation

### Initial structures

NPRS1 sequence was retrieved in the FASTA format form the UniPort database (UniProtKB —Q6W5P4) [16]. The I-TASSER server was utilized to construct the initial structure of NSPR1 [17, 18]. The crystal structure of bovine rhodopsin from the Protein Data Bank (PDB: 1U19:A) has been used as a template for the construction [19]. All default parameters without any constraints were employed. The best-predicted structure (C-score = 0.62) was selected for the simulation. UCSF Chimera (version 1.13.1) was applied to mutate the 107th amino acid in the structure from asparagine to isoleucine (Asn107Ile) [20].

### System setup

Protein structures were viewed and examined with Visual Molecular Dynamics (VMD) [21]. VMD psfgen plugin was used to generate a dynamics-ready psf and PDB files from a raw PDBs [21]. Both structures have been inserted palmitoyl-oleoyl-phosphatidyl-choline (POPC) with all overlapping lipid molecules removed using the Membrane plugin and solvated in a water box using the Solvate plugin in VMD [21].

### Molecular dynamics (MD) simulations

Molecular dynamics (MD) simulations were made using NAMD2.9 [22]. The CHARMM27 parameter was used for the protein and the POPC lipids [23]. Simulations of both structures started with equilibration of the lipid tails. With all other atoms fixed, the lipid tails were energy minimized for 1000 steps using the conjugate gradient algorithm for 0.5 ns at 300 K. The systems were further equilibrated at 300 K and constant pressure with harmonic position restraints applied to the protein atoms for 0.5 ns. The systems were further equilibrated at 310 K and constant pressure with all atoms unrestrained for 0.5 ns. At this point, the system volume was found to decrease; suggesting that water molecules, ions, and lipids were well equilibrated surrounding the protein structures. The final production run for both systems was performed at 1 atm pressure and 310 K for 2 ns with a constant ratio constraint applied on the lipid bilayer.

### Molecular dynamics simulations result analysis

The R statistical language and R studio were used for the analysis of results [24, 25]. Bio3D R package was used to calculate and generate RMSD, RMSF graphs [26]. The R package ggplot2 was used to generate histograms [27].

### Pocket size calculation

The size of the pocket inside NPSR1 for NPS binding was calculated using CASTp software.

## Statistical analyses

Two-way analysis of variance ANOVA was applied to analyze biochemical tests data means of obese and controls. Chi-square test, Z-test, dominance, recessive model and additive tests, odd ratio, Fisher exact were performed to analyze genotype and allele frequencies. The statistical significance level was set at $p < 0.05$. Andrew F. Hayes Moderation effect analysis was performed using process Version 3.5 to analyze the effect of obesity as independent variable directly on NPS concentration considering genotype as moderator.

## Results

### Anthropometric and lipid profile of participants

A total 116 participants with similar age and sex were included in this study, out of which 49% were obese, while 51% were healthy individuals (Table 1). This study shows that the mean values of BMI, cholesterol, triglycerides and LDL were significantly ($p<0.05$) higher in the obese group while NPS concentration were found significantly ($P<0.001$) lower as compared to control group (Table 1). However, blood glucose (fasting), HDL concentrations were found non-significant ($p>0.05$) different in obese and control group (Fig 1).

### Genotyping

**Genetic models.** PCR-RFLP results confirmed by Sanger sequencing technique (S1 Fig) revealed, 28 AA and 29 TT+ AT in obese while 40 AA and 19 TT+AT in healthy individuals. The reported genotypes single nucleotide NPSR1 gene rs324981(A>T) polymorphisms were in Hardy-Weinberg equilibrium (HWE) ($p2 + 2pq + q2 = 1$) and the observed and expected genotype frequencies were found significantly ($p = 0.005$) different through Chi-square statistic. The current study has shown a significant ($p = 0.04$) difference of NPSR1 Asn107Ile (rs324981) geneotype frequencies of between obese and controls. The analyses of NPSR1 rs324981 (A>T) polymorphism find a significant association with obesity in Dominant genetic models (AA Genotype vs AT + TT Genotype) (Table 2). However, non-significant results of Recessive genetic model (AA +AT Genotype vs TT Genotype) with their significant ($p = 0.04$) Chi- Square and Z-test values are shown in Table 2. The 'T' allele frequency was high in obese (38%) than in healthy individuals (25%) but insignificant as shown (OR = 1.83, 95%CI, 0.829–4.125, $p = 0.06$) (Table 2). Moreover, in additive model (A vs T) Z-test and chi-square found a non-significant difference in rs324981 allele frequencies between the obese and the control group (Table 2).

**Table 1. Anthropometric and lipid profile of participants.**

| Parameters | Obese | Controls | 95% CI of diff. | P value |
|---|---|---|---|---|
| Age | 35.8 | 32.4 | - | p>0.12 |
| BMI Kg/m2 | 32.90* | 21.83* | -20.27 to -1.872 | P<0.01 |
| Blood Sugar (mg/dl) | 98.08 | 95.12 | -12.16 to 6.239 | P > 0.05 |
| Total cholesterol (mg/dl) | 196.1** | 106.3** | -98.95 to -80.55 | P<0.001 |
| Triglycerides (mg/dl) | 76.79** | 51.63** | -34.35 to -15.95 | P<0.001 |
| HDL (mg/dl) | 35.94 | 42.96 | -2.182 to 16.22 | P > 0.05 |
| LDL (mg/dl) | 97.35** | 56.94** | -49.60 to -31.20 | P<0.001 |
| NPS (pg/ml) | 99.58** | 195.4** | 90.05 to 108.5 | P<0.001 |

Results of Two way Anova (Obese. Ctrl Age, BMI, Glucose, Cholesterol, Triglycerides, HDL, LDL, NPS conf. level = 0.95) * Indicates p < 0.05.

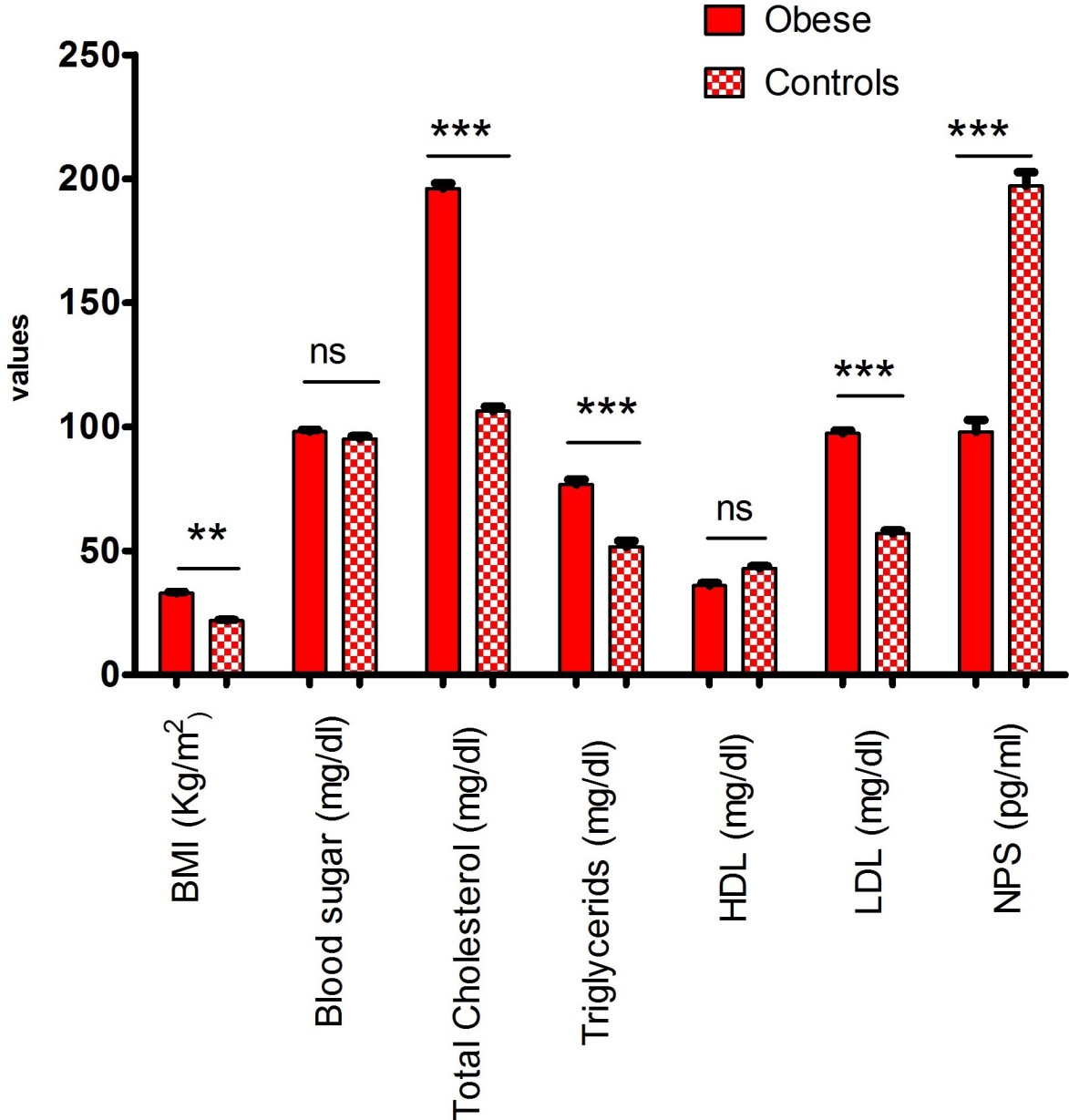

**Fig 1. Lipid profile and anthropometric parameters vary significantly between obese and control groups.**

**Table 2. Genotypic distribution of NPSR1 (rs324981, A>T) in normal controls (n = 59) and obese (n = 57) Pakistani individuals.**

|  | Model | OR(95% CI)P-Value | $X^2$(p-value) | Z-test(P-value) |
|---|---|---|---|---|
| **rs324981** |  |  |  |  |
|  | Dominant | 2.165(1.019–4.70)0.04 | 4.131(0.04) | -2.041(0.04) |
|  | Recessive | 0.629(0.245–1.571)0.36 | 4.131(0.04) | 2.041(0.04) |
|  | Additive | 1.83(0.829–4.125)0.06 | 2.31(0.128) | -1.52 (0.12) |

$p$-Values<0.05 were considered significant. NS: non-significant.

Moreover, we applied two way ANOVA model to compare the means of independent variables obesity and genotype against the dependent variables Age, NPS and lipid profile parameters both in obese and control groups.

In this study we found the means for NPS and triglycerides were significantly (p = 0.0001) different for different genotypes in obese and control group. Furthermore, means for Cholesterol, BMI, HDL, blood sugar and LDL are significantly different in obese and control group (p<0.01). Interestingly only in the case of triglycerides there is an interaction between the independent variables obesity and genotype (Fig 2, Table 3).

We found a statistically significant difference in average NPS by both SNP (f(2) = 26.0613, p <5.435e-10) and by obesity (f(1) = 186.6127, p<2.2e-16), though the interaction between these terms was not significant. A Tukey post-hoc test revealed significant pairwise differences between genotype "AA" and genotype "AT" (diff = 47.930945), between "AA" and genotype TT" (diff = 47.930945). While no significant difference was found between "AT" and genotype TT" (diff = -2.531281). Moreover, We found a statistically significant difference in average Triglycerides by both SNP (f(2) = 4.7670, p = 0.01034) and by obesity (f(1) = 399.2172, p<2.2e-16); however the interaction between these terms was not significant. NPS is the only dependent variables that show a significant difference in both SNP and obesity.

**Correlation analysis.** The data showed a statistically significant ($p<0.0001$) difference of mean NPS serum level 99.58 ±34.43 pg/mL in obese compared to controls 195.49 ±38.87 pg/ml (Fig 3). There was a statistically significant negative correlation ($r^2$ = 0.73, $p$ = 0.0001) between BMI score of all participants and NPS serum levels (Fig 3).

**Moderation effect analysis.** NPS concentrations were significantly (p = 0.000) moderated by NPSR1 gene Asn107Ile polymorphism with increasing obesity (Fig 4). Furthermore moderation effect analyses revealed a more negative effect of obesity on NPS concentration with AT +TT and AA genotype as shown in (S1 Table).

**Molecular simulation effect.** Molecular simulations were performed to understand the effect of the mutation on the interaction of NPSR1 and NPS. The WT trajectory is shown in black, and the variant Asn107Ile trajectory is shown in red (Fig 5A and 5B). Fig 5C and 5D showed the WT and variant molecular structure trajectories have difference in the structures as measured by the RMSD (root mean squared deviation) (Fig 5C). The average RMSD is shown by the blue dashed line with the WT above the variant. This distribution of the RMSD of the population of structures obtained during the final 2ns of the simulation is shown for the WT (red) and variant ASN107Ile (blue) in Fig 5C and 5D. The differences in RMSD distributions for the population of structures are more pronounced when isolating residues 100–330. The simulations predict that the binding of NPS to NPSR1 differs in the WT vs. the variant Asn107Ile. It is interesting to note that the difference in RMSF between WT and variant is greatest with residue 198 suggesting its importance in NPS binding. S2 Fig shows the FCC scores for the WT and variant. The higher the score the closer the interaction between NPS and NPSR1. Note that the values in the WT are greater. In fact, in cluster 1 (red—circled) which includes residues (100–330), the WT has a value slightly above 0.6 while the Asn107Ile variant is zero, suggesting a reduced binding affinity of NPS for the Asn107Ile variant of NPSR1.

The three-dimensional structure of wild-type and Asn107Ile NPSR1 obtained by molecular simulation from Fig 5A and 5B. The relative "pocket sizes" for binding of NPS to NPRS1 WT and NPRS1 Asn107Ile calculate by CASTp were 732.885A and 449.454A, respectively (Fig 5E and 5F). These predictions suggest that NPS binds the Asn107Ile variant less strongly than NPS binds to the wild-type due to the changes in the NPSR1 structure.

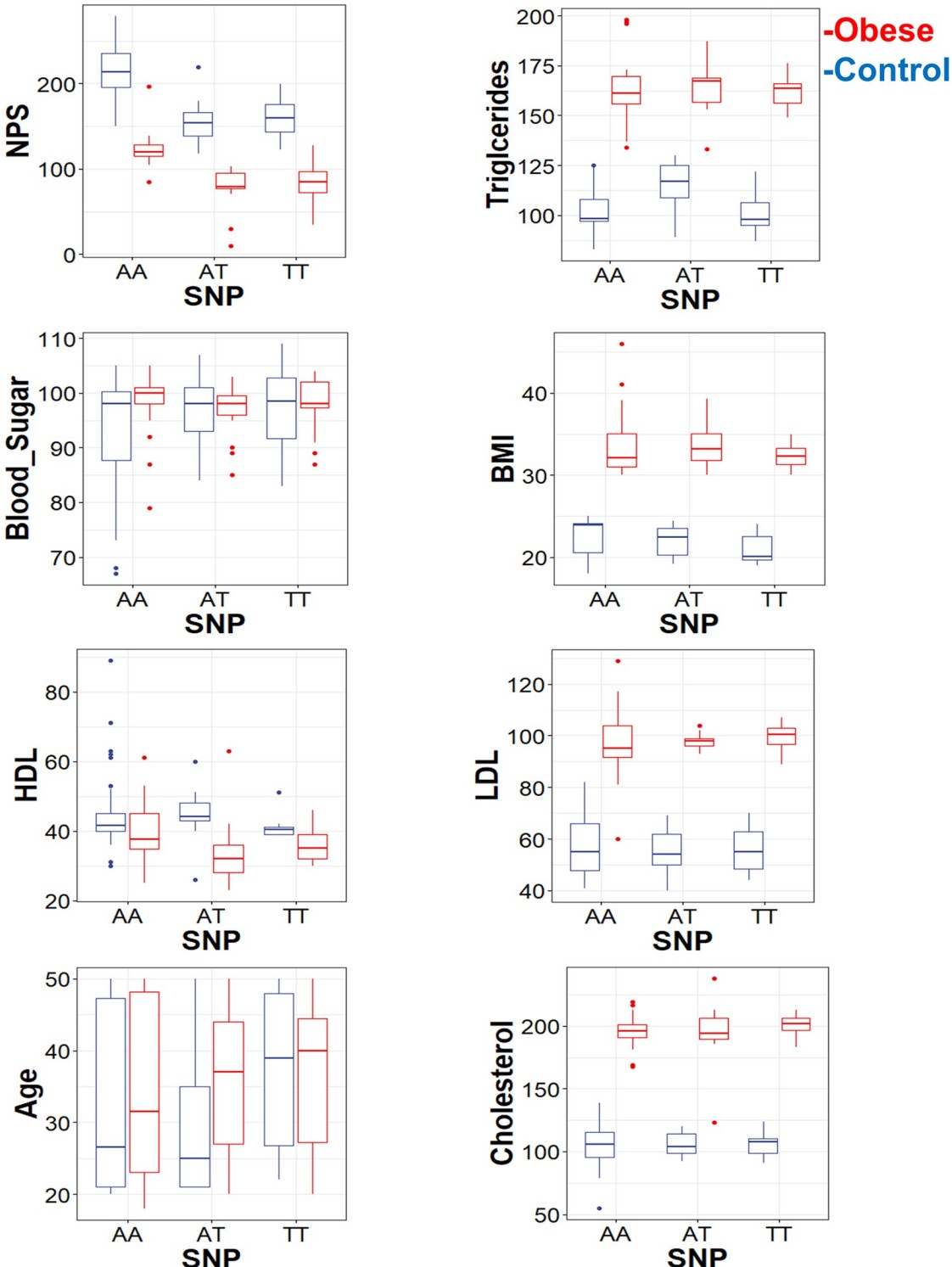

**Fig 2. Two-way ANOVA analysis with obesity and genotype as the independent variables and NPS, triglycerides, blood sugar, BMI, HDL, LDL, age, and cholesterol are significantly p = 0.0001 different for different genotypes and for obese vs non-obese subject.**

**Table 3. Comparison of independent variables obesity and genotype against dependent variables (NPS, triglycerides, blood sugar, BMI, HDL, LDL, age and cholesterol) by two-way analysis.**

| Variable | p-value | Tukey HSD p-value |
|---|---|---|
| NPS | obesity–$<2.2\times10^{-16}$ | AT-AA– 0.0 |
|  | genotype– $5.4\times10^{-10}$ | TT-AA– 0.0 |
|  | obesity:genotype– 0.61 | TT-AT– 0.94 |
| Triglycerides | obesity–$<2.0\times10^{-16}$ | AT-AA– 0.072 |
|  | genotype–$<2.0\times10^{-16}$ | TT-AA– 0.70 |
|  | obesity:genotype– 0.01 | TT-AT– 0.92 |
| Blood Sugar | obesity– 0.004 | AT-AA– 0.94 |
|  | genotype–.019 | TT-AA– 0.098 |
|  | obesity:genotype– 0.19 | TT-AT– 0.40 |
| BMI | obesity–$<2.0\times10^{-16}$ | AT-AA– 0.88 |
|  | genotype– 0.23 | TT-AA– 0.098 |
|  | obesity:genotype– 0.90 | TT-AT– 0.40 |
| HDL | obesity– 0.008 | AT-AA– 0.28 |
|  | genotype– 0.49 | TT-AA –0.35 |
|  | obesity:genotype– 0.47 | TT-AT– 0.99 |
| LDL | obesity–$<2.0\times10^{-16}$ | AT-AA– 0.96 |
|  | genotype– 0.84 | TT-AA– 0.97 |
|  | obesity:genotype– 0.72 | TT-AT– 0.91 |
| Age | obesity– 0.23 | AT-AA– 0.99 |
|  | genotype– 0.33 | TT-AA– 0.53 |
|  | obesity:genotype– 0.58 | TT-AT– 0.64 |
| Cholesterol | obesity–$<2.0\times10^{-16}$ | AT-AA– 0.99 |
|  | genotype– 0.99 | TT-AA– 0.70 |
|  | obesity:genotype– 0.79 | TT-AT– 0.73 |

## Discussion

Worldwide over 650 million individuals are obese. The World Health Organization (WHO) has declared obesity as an epidemic, especially in low and middle-income countries that are more vulnerable with the double burden of disease coupled with obesity. i.e., Pakistan consumes 0.71% of total health care expenditure directly and 11.7% indirectly to cure obesity and its co-morbidities. The risk for this non-contagious disease increases with increases in BMI [1, 28]. This study revealed a negative correlation of NPS concentration in serum versus BMI which decreased significantly with increasing BMI scores. This may be because of the underrepresentation of the AA genotype in obese Pakistani participants. Previous studies specify the possible defensive role of AA genotype against obesity with relatively less frequent occurrence in the Asian Population [8]. It has a protective role in asthma, that helped to reduce airway hyper-responsiveness as reported for the Chinese population [29] while the under-representation of AA genotype was found associated with panic disorder in the Japanese population [8]. Additionally, participants with AA genotype have equally good sociability characteristics and better adaptability to the environment with a minimum level of anxiety and depression [8, 30]. Mechanistically, negative correlation of NPS concentration in serum with BMI may be because of reduced neuropeptide binding of A allele (Asn 107) at the receptor site compared to the T allele (107Ile) [13]. The T allele has been reported with increased neural stress in the Estonian males [26].

There was a significant difference between AA and AT+TT genotype frequencies of NPSR1 gene between the obese and control group. However, NPS concentrations in serum varied

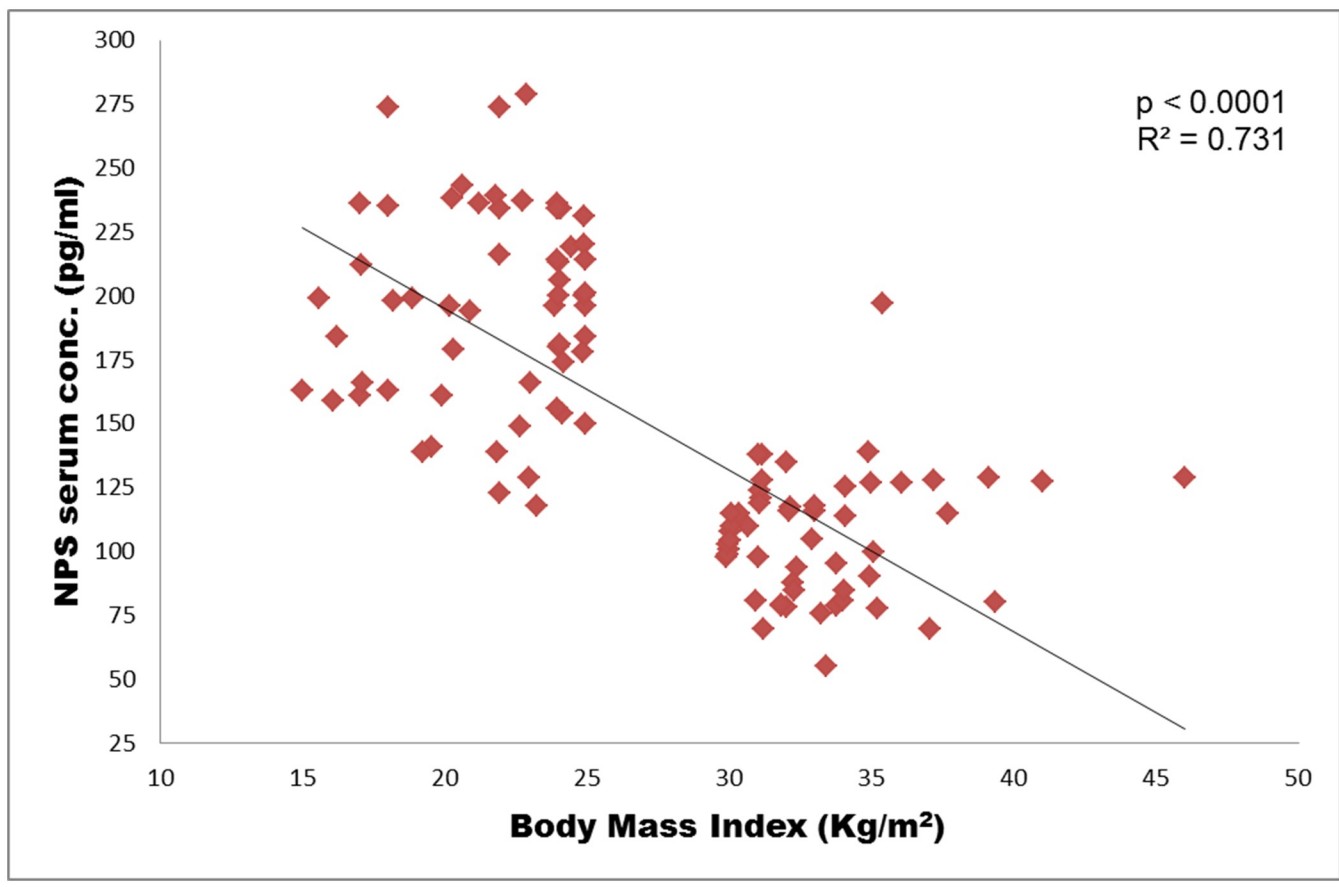

**Fig 3. Pearson correlation analysis of NPS serum concentration (pg/ml) versus BMI of all the samples (n = 116).** ($r^2$ = 0.73, p = 0.0001).

significantly with genotype between the obese and control group. Furthermore, genotype as moderator in moderation effect analysis, negatively affected the NPS concentration with increasing obesity. These results suggest that presence of T allele is not only the single factor responsible for the increase of NPSR1 sensitivity to NPS as reported earlier [13], besides the fact TT+AT genotype was frequently observed in Pakistani obese male participants, very similar to previous Asian and German population studies [8, 29]. Similarly, Frequency of risk allele T was recorded higher (38%) in obese Pakistani participants as compared to their controls (25%). Furthermore, NPSR1 Asn107Ile variant (AT+TT) genotype distribution was found to be associated with obesity in the Pakistani population. Similarly reduced cognitive stress in healthy Chinese has been found associated with the TT genotype of NPSR1 rs324981 [31–33]. This may because of environment-dependent expression of NPSR1 gene, as revealed earlier in malnourished individuals with TT genotype were found highly sensitive against depression and anxiety [11] and that may be the reason for the association of (rs324981) T allele with anxiety [34] impulsivity [35] sleep disorder [36] and panic disorder [37, 38].

In this study, we revealed a significantly decreased level of NPS concentration in obese individuals dependent of genotype and allele frequency. That may because of reduced signaling transduction of T allele (107Ile) at (rs324981) of NPSR1 gene with its attenuated binding affinity to NPS and altered biological response at receptor site [12, 39, 40]. Therefore, NPS receptor the NPSR1 with elevated agonistic response possibly may contribute towards hyper-stimulation and neurotransmission which ultimately may result in the pathophysiology of obesity in

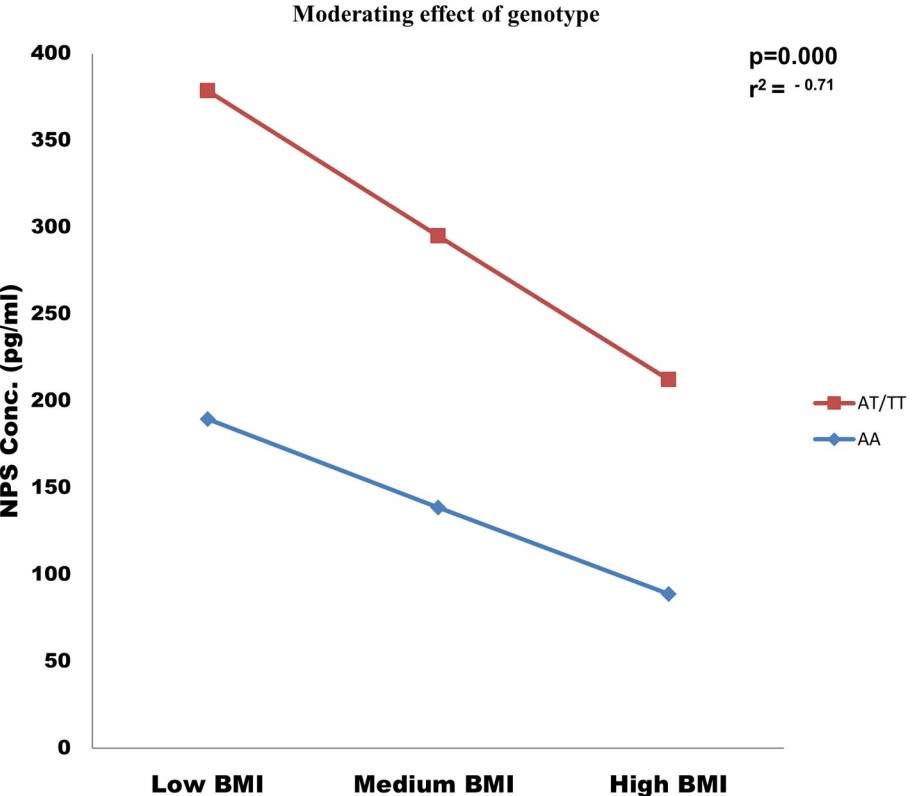

**Fig 4. Moderation effect of analysis, genotype (AT+TT and AA) negatively moderated the effect of obesity on NPS concentration.**

the Pakistani population, but still, further investigations with larger data sets are required to evaluate in the context of NPS-NPSR1 interaction.

This study is first to report the significant association of NPSR1 (Asn107Ile rs324981) polymorphism with obesity in the Pakistani obese male individuals. Smaller data sets can be a potential limitation of this investigation but its significance in terms of outcomes for the scientific community cannot be undermined. Our results NPSR1 A/T polymorphisms provides further supportive evidence in the association of obesity. However, in this study, we observed that the AA genotype was significantly associated with healthy male individuals in Pakistan.

We observed a reduced level of NPS concentration in obese individuals with weaker binding affinity to the NPSR1 Asn107Ile variant. That might have contributed have contributed towards the pathophysiology of obesity In addition, the NPS-NPSR1 system on downstream accelerates MAPK activity and fatty acid oxidation to protect organisms against diet-induced obesity [41]. Moreover, previous experimental studies data suggest the decreased activation of NPSR1-NPS signaling can lead to obesity [42]. Furthermore, NPS-NPSR1 activation results in the increased expression of cholecystokinin (CCK), vasoactive intestinal peptide (VIP), and neuropeptide Y [42]. CKK and VIP are responsible for the pancreatic secretion and nutrient absorption in the intestine region collectively can modulate PYY release [43]. Neuropeptide Y regulates food intake decreasing intestinal motility giving the feeling of satiety [44]. Furthermore, the NPSR1 Asn107Ile variant has been shown to increase cortisol levels in European subjects [32, 45]. Therefore, reduced NPSR1-NPS signaling activates pathways that lead to weight gain.

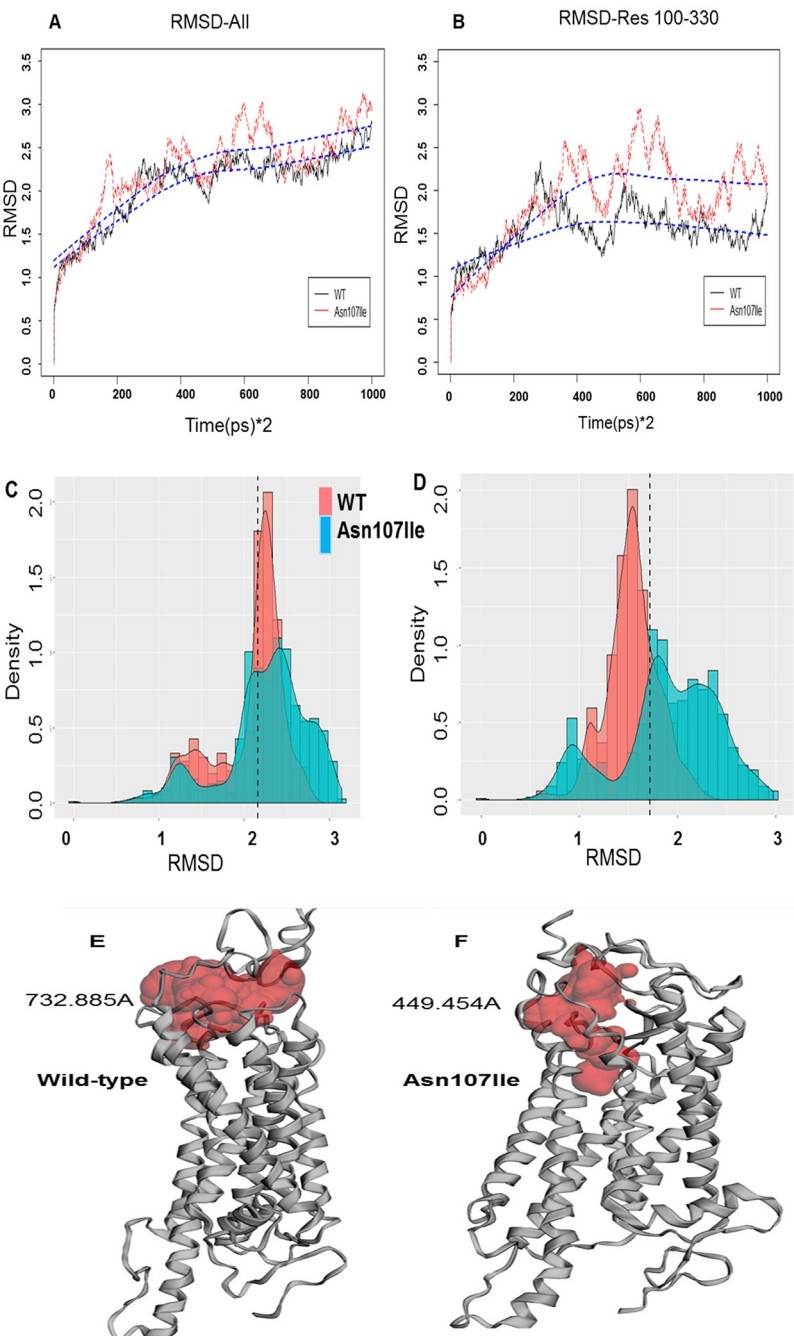

**Fig 5.** The root means squared difference (RMSD) of both WT and Asn107Ile in (A) all residues, and (B) residues 100–300 for each frame (each frame is 2 ps of a simulation). There is a clear divergence between the two structures when comparing residues 100–330. (C) RMSD Histogram for NPRS1 WT, and NPRS1 Asn107Ile obtained during the final 2 ns of the simulation. (D) The changes in RMSD distributions s more noticeable when isolating residues 100–330. (E) Molecular structure (gray) of E. wild-type and (F) Asn107Ile variant NPSR1 obtained by molecular simulation with the pocket calculated by CASTp (red). The pocket-size is 732.885 A for WT and 449.454 for Asn107Ile.

This study also provides insight into the molecular mechanism defining the difference between WT and variant Asn107Ile. The modeling suggests a reduced binding affinity of NPSR1 for NPS in variant Asn107Ile compared to WT similar to experimental data findings [41]. The simulation studies suggest that the amino acid residue 198 is important in the

interaction between NPS and NPSR1. The variant Asn107Ile displays increase fluctuation in molecular structure most significantly at residue 198 suggesting its importance in NPS binding to NPSR1. This predicts that the mutation at Asn107 might exert its effect through a distant residue. Experimental findings support the importance of residue 198; the NPSR1 variant Phe197Cys shows no binding of NPS [41]. Further studies are needed to test the predicted critical role of residue 198 and the involvement of the other predicted residues involved in this interaction as these might play an important pharmacological role.

## Supporting information

**S1 Fig. Verification of the NPSR1 A>T (rs324981) SNP between (A) obese (n = 57) and (B) healthy (n = 59) by Sanger sequencing.**
(TIF)

**S2 Fig. Shows the FCC scores for the WT and variant.** The higher the score the closer the interaction between NPS and NPSR1. In cluster 1 (red—circled) which includes residues (100–330).
(TIF)

**S1 Table. Moderation effect analysis, when genotypes act as a moderator while obesity is independent variable and NPS is response variable.**
(DOCX)

**S1 File.**
(ZIP)

## Acknowledgments

The authors appreciate the technical supports of Alpha Genomics and would like to express thanks. The authors acknowledge the patient recruitment support from doctors of PESSI Hospital Islamabad and sincerely thank the participants for this study.

## Author Contributions

**Conceptualization:** Pasha Ghazal, Habib Bokhari.

**Data curation:** Aftab Ahmad.

**Formal analysis:** Aftab Ahmad, Malik Waqar Ahmed.

**Investigation:** Aftab Ahmad.

**Methodology:** Aftab Ahmad, Fahad Almsned.

**Project administration:** Pasha Ghazal.

**Resources:** Pasha Ghazal, Habib Bokhari.

**Software:** Aftab Ahmad, Fahad Almsned, Malik Waqar Ahmed, M. Saleet Jafri.

**Supervision:** Pasha Ghazal, M. Saleet Jafri, Habib Bokhari.

**Validation:** Aftab Ahmad, M. Saleet Jafri.

**Visualization:** Fahad Almsned, M. Saleet Jafri.

**Writing – original draft:** Aftab Ahmad.

**Writing – review & editing:** Aftab Ahmad, Fahad Almsned, M. Saleet Jafri.

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
