## [Decision Letter · Decision Letter 0]

20 Mar 2020

PONE-D-19-35919

Neuropeptide S receptor gene Asn107 polymorphism in obese male individuals in Pakistan

PLOS ONE

Dear Dr. Bokhari,

Thank you for submitting your manuscript to PLOS ONE. After careful consideration, we feel that it has merit but does not fully meet PLOS ONE’s publication criteria as it currently stands. Therefore, we invite you to submit a revised version of the manuscript that addresses the points raised during the review process.

We would appreciate receiving your revised manuscript by May 04 2020 11:59PM. To enhance the reproducibility of your results, we recommend that if applicable you deposit your laboratory protocols in protocols.io, where a protocol can be assigned its own identifier (DOI) such that it can be cited independently in the future. For instructions see: http://journals.plos.org/plosone/s/submission-guidelines#loc-laboratory-protocols

We look forward to receiving your revised manuscript.

Kind regards,

Zezhi Li, Ph.D., M.D.

Academic Editor

PLOS ONE

Additional Editor Comments (if provided):

The sample size is too small, so the authors should provide the statistical power. In addition, it should be discussed in the limitation.

Journal Requirements:

2. Please upload a copy of Figure 8, to which you refer in your text on page 16. If the figure is no longer to be included as part of the submission please remove all reference to it within the text.

3. Please upload a new copy of Figure S8 as the detail is not clear. Please follow the link for more information: http://blogs.PLOS.org/everyone/2011/05/10/how-to-check-your-manuscript-image-quality-in-editorial-manager/

Reviewers' comments:

Reviewer's Responses to Questions

**Comments to the Author**

1. Is the manuscript technically sound, and do the data support the conclusions?

Reviewer #1: No

Reviewer #2: No

2. Has the statistical analysis been performed appropriately and rigorously? 

Reviewer #1: Yes

Reviewer #2: No

3. Have the authors made all data underlying the findings in their manuscript fully available?

Reviewer #1: Yes

Reviewer #2: Yes

4. Is the manuscript presented in an intelligible fashion and written in standard English?

Reviewer #1: No

Reviewer #2: Yes

5. Review Comments to the Author

Reviewer #1: I read carefully version of the article titled “Neuropeptide S receptor gene Asn107 polymorphism in obese male individuals in Pakistan”

I have several major comments listed below ---

1. Neuropeptide S receptor (NPSR1) gene might have many SNPs, the author should describe the possible reason why they choose rs324981 to study.

2. The figures are too obscure to be seen. The authors should provide clearer graph.

3. The language and paper structure is too obscure to understand. Such as “clinical picture “. What is the meaning of clinical picture?

4. The sample size is too small for SNP polymorphism study.

Reviewer #2: 1. For a genetic study, the sample size is too small.

2. According to the data given in the manuscript, the result of the chi-square test for genotypes is incorrect. The chi-square value should be 4.251, which does not reach a significant level (p=0.119), and the z-test was not significant either. It is not clear how chi-square=4.131 (p=0.04) is calculated. Therefore, in my opinion, there are no significant results in this study.

Here is my chi-square test results:

obesity * genotype Crosstabulation

Count

genotype Total

AA TT AT

obesity Yes 28a 14a 15a 57

No 40a 10a 9a 59

Total 68 24 24 116

Each subscript letter denotes a subset of genotype categories whose column proportions do not differ significantly from each other at the .05 level.

Chi-Square Tests

Value df Asymptotic Significance (2-sided)

Pearson Chi-Square 4.251a 2 .119

Likelihood Ratio 4.280 2 .118

Linear-by-Linear Association 3.914 1 .048

N of Valid Cases 116

a 0 cells (0.0%) have expected count less than 5. The minimum expected count is 11.79.

6. PLOS authors have the option to publish the peer review history of their article (what does this mean?). If published, this will include your full peer review and any attached files.

Reviewer #1: No

Reviewer #2: Yes: Jiesi Wang

---

## [Author Response · Author response to Decision Letter 0]

21 May 2020

Response Letter addressing “Revision”

Dear Editor PLOS ONE,

With Reference ;

PONE-D-19-35919

Neuropeptide S receptor gene Asn107 polymorphism in obese male individuals in Pakistan

The Manuscript Revision comments & Response here for your consideration.The authors appreciate the time the reviewers have invested in reviewing our manuscript carefully and for their valuable input which has significantly enhanced the impact of this manuscript. Following are the responses to reviewer comments and subsequent changes in the revised manuscript.

The specific comments and their responses are mentioned below.

Reviewer 1

Comment # 1. Neuropeptide S receptor (NPSR1) gene might have many SNPs, the author should describe the possible reason why they choose rs324981 to study.

Response. NPS-NPSR1 interaction have been discussed in discussion section of manscript, this system on downstream accelerates MAPK activity and fatty acid oxidation to protect organisms against diet-induced obesity (Reinscheid et al., 2005; Anedda F., et al 2011). In cell models, the change of Asn(107) to Ile(107) results in 10-fold increase in NPS-mediated intracellular signaling (Bernier et al., 2006; Anedda et al., 2011). A nonsynonymous SNP at rs324981 is more functional gene variation among the well-known NPSR1 SNPs rs324987, rs324957, rs323920, rs324396 and rs323922 (Yan Feng, et al., 2006). GWAS have found an association of rs324981 with major obesity linked comorbidities like IBD, depression, anxiety, panic disorder, sleep and rest in diverse populations but still true causative variations remain to be identified (Gottlieb DJ., et al 2007; Reinscheid, 2008; Domschke et al., 2011; Lennertz L. et al., 2013).

Comment # 2. The figures are too obscure to be seen. The authors should provide clearer graph.

Response: We have revisited the all figures addressed the issues according to PLOS ONE figures guidelines https://journals.plos.org/plosone/s/figures. 

Comment # 3. The language and paper structure is too obscure to understand. Such as “clinical picture “. What is the meaning of clinical picture?

Response; We have revisited the complete manuscript considering these critical comments and addressed them accordingly. There are number of sentences which have been rephrased for better expression. Moreover, language and grammar was thoroughly checked from abstract to conclusions sections and all the necessary technical expression have been amended accordingly. 

a) Language and grammar revised by 

Habib Bokhari, PhD

Commonwealth Scholar & Fellow (LSHTM, UK)

Fulbright Fellow (Perelman School of Medicine, UPENN, USA)

Professor, Department of Biosciences COMSATS University, Islamabad, Pakistan

Office: +92-51-250-1223; FAX: +92-51-444-2805 Mobile: +92-300-512-7684

E-mail: habib@comsats.edu.pk

b) A copy of manuscript showing track changes has been uploaded.

c) A clean copy of the edited manuscript has been uploaded as the new manuscript file.

Comment # 4. The sample size is too small for SNP polymorphism study.

Response; This study has potential limitations because of limited resources availability, time constraint and stringent inclusion/exclusion criteria for both case & controls. However, the data does have statistical significance as described in the next comment. We have added discussion about this topic to the discussion.

Reviewer # 2: 

Comment # 1. For a genetic study, the sample size is too small.

Response; This study has potential limitations because of limited resources availability, time constraint and stringent inclusion/exclusion criteria for both case & controls. However, the data does have statistical significance as described in the next comment. We have added discussion about this topic to the discussion.

Comment # 2. According to the data given in the manuscript, the result of the chi-square test for genotypes is incorrect. The chi-square value should be 4.251, which does not reach a significant level (p=0.119), and the z-test was not significant either. It is not clear how chi-square=4.131 (p=0.04) is calculated. Therefore, in my opinion, there are no significant results in this study.

Response; Actually we applied the Dominant model considering “T” as a dominant allele (Feifei Zhao et al., 2016). We have reordered data table accordingly.

 Obese Control Row Total 

T 29(60.42%) 19(39.58%) 48(100%)

A 28(41.18%) 40(58.82%) 68(100%)

Column Total 57 58 116

The chi-square statistic is 4.1677. The p-value is .041201. Significant at p < .05.

Additional Editor Comments 

The sample size is too small, so the authors should provide the statistical power. In addition, it should be discussed in the limitation.

Response; This study has potential limitations because of limited resources availability, time constraint and stringent inclusion/exclusion criteria for both case & controls. However, the data does have statistical significance as described in the next comment. We have added discussion about this topic to the discussion.

Comment # 1. Please ensure that your manuscript meets PLOS ONE's style requirements, including those for file naming. 

Response; Thanks for your suggestion, We have revised the all file names and style to conform closely to PLOS ONE's requirements.

Comment # 2. Please upload a copy of Figure 8, to which you refer in your text on page 16. If the figure is no longer to be included as part of the submission please remove all reference to it within the text.

Response. Yes, figure reference has been removed from the text and figure 8 is no longer part of submission.

Comment # 3. Please upload a new copy of Figure S8 as the detail is not clear.

 Response. Thanks for your input, a new copy of Fig S8 labeled as Fig S7 with clear details has been uploaded.

---

## [Decision Letter · Decision Letter 1]

24 Jun 2020

PONE-D-19-35919R1

Neuropeptide S receptor gene Asn107 polymorphism in obese male individuals in Pakistan

PLOS ONE

Dear Dr. Bokhari,

Thank you for submitting your manuscript to PLOS ONE. After careful consideration, we feel that it has merit but does not fully meet PLOS ONE’s publication criteria as it currently stands. Therefore, we invite you to submit a revised version of the manuscript that addresses the points raised during the review process.

We look forward to receiving your revised manuscript.

Kind regards,

Zezhi Li, Ph.D., M.D.

Academic Editor

PLOS ONE

Reviewers' comments:

Reviewer's Responses to Questions

**Comments to the Author**

1. If the authors have adequately addressed your comments raised in a previous round of review and you feel that this manuscript is now acceptable for publication, you may indicate that here to bypass the “Comments to the Author” section, enter your conflict of interest statement in the “Confidential to Editor” section, and submit your "Accept" recommendation.

Reviewer #2: (No Response)

2. Is the manuscript technically sound, and do the data support the conclusions?

Reviewer #2: No

3. Has the statistical analysis been performed appropriately and rigorously? 

Reviewer #2: No

4. Have the authors made all data underlying the findings in their manuscript fully available?

Reviewer #2: No

5. Is the manuscript presented in an intelligible fashion and written in standard English?

Reviewer #2: Yes

6. Review Comments to the Author

Reviewer #2: This revision has some improvement compare to the original one, but the accuracy of the data in this manuscript is still my biggest concern. The following comments are some of the main issues that I think the author should explain or improve. And the corresponding conclusions and discussions also need to be revised, because many conclusions are not supported by data or statistics.

Comment1: The genotype of Asn107lle in this manuscript did not conform to the Hardy Weinberg Equation, this should be reported in the article. And the deviation of HWE may due to the small sample size, but for me, it much more like there have genotype errors, especially by using a PCR-RFLP method. The authors said they have confirmed the results by Sanger sequencing, how much samples have been confirmed, especially for Heterozygotes and Rare Homozygotes?

Comment2: Some data in the table1 is confusing. For Glucose, obese vs healthy is 94.5 ± 9.1 vs 94.5± 9.2, but p<0.0001, that won’t be possible. The Total cholesterol in control is 105.5 ± 13.6, but the Triglycerides is 104.5± 11.9, it was also impossible, because the Triglycerides is normally about 30% of the Total cholesterol, in this table, it is more than 90%. I would not report the data if I can’t confirm the accuracy.

Comment3: According to the date of age, two groups was not age-matched, but the difference of age has not reached a significant level (p>0.12) only. Because the age may have a large influence on BMI, they should consider to be covariates in the correlation analysis or ANOVA analysis.

Comment4: If the authors choose the Dominant model in genotype analysis, it also should be used in the whole manuscript.

Comment5: Two-way ANOVA used in table1? What is the second independent variable besides obesity? Indeed, a two-way ANOVA with obesity and genotype of Asn107lle should be performed with the dependent variables including BMI, NPS level, and other serum indexes in table1 (if they are corrected). There is no need to compare the NPS level between obesity and control in different genotypes unless there has a significant interaction of obesity and genotype on the NPS level.

Comment6: The authors reported rs=0.884, p=0.0004 in the result (what is the meaning of rs?), but in the figure, they show r2=0.1797, p<0.0001? And it also seemed there did not have 116 plots on the figure, please confirmed and provided a more cleared one.

Comment7: Molecular simulation can only provide a hypothesis that the SNP may influence the interaction of NPS and Asn107lle. But no real data supported the interaction was affected, and furthermore, this effect has an influence on obesity. Indeed, the authors should perform a moderation effect analysis using the genotype as a moderating variable or perform a mediation effect analysis using the NPS as a mediating variable, these analyses can check if the interaction of genotypes and NPS has an effect on obesity.

Comment8: Too many figures. Figure 1 does not need to be provided in the main text; Figure 2 and 3 need to be re-figured according to a new two-way ANOVA analysis; Figure5-7 can be put together into one figure.

Comment9: The simulation is just a prediction; I would not write three long paragraphs to report these results, even longer than the results from real data.

Comment10: The RESULTS may be better presented with separate paragraphs accompanied by sub-titles.

7. PLOS authors have the option to publish the peer review history of their article (what does this mean?). If published, this will include your full peer review and any attached files.

Reviewer #2: **Yes: **Wang Jiesi

---

## [Author Response · Author response to Decision Letter 1]

11 Aug 2020

Response Letter addressing “Revision”

Dear Editor PLOS ONE,

With Reference ;

PONE-D-19-35919

Neuropeptide S receptor gene Asn107 polymorphism in obese male individuals in Pakistan

The Manuscript Revision comments & Response here for your consideration. The authors appreciate the time the reviewers have invested in reviewing our manuscript carefully and for their valuable input which has significantly enhanced the impact of this manuscript. Following are the responses to reviewer comments and subsequent changes in the revised manuscript.

The specific comments and their responses are mentioned below.

Reviewer 2

Comment # 1. Comment1: The genotype of Asn107lle in this manuscript did not conform to the Hardy Weinberg Equation, this should be reported in the article. And the deviation of HWE may due to the small sample size, but for me, it much more like there have genotype errors, especially by using a PCR-RFLP method. The authors said they have confirmed the results by Sanger sequencing, how much samples have been confirmed, especially for Heterozygotes and Rare Homozygotes?

Response. Yes, genotype of all 116 samples reported in this manuscript was confirmed by Sanger sequencing rather than relying on PCR-RFLP method, and sequencing results are submitted here as a “supplementary Sanger sequencing file”. Furthermore this genotype data follows the Hardy Weinberg Equation, calculations are shown below. 

 Genotype Allelic Frequency

AA TT AT A T

68 24 24 0.69 0.31

f(T)= (AT)+2 X (TT)/2 X (TT) + 2 X (24) + 2 X (AA)

 =24+ 2(24)/2(24)+2(24)+2(68)

 = 72/232

 = 0.31

f(A)= (AT)+2 X (AA)/2 X (TT) + 2 X (AT) + 2 X (AA)

 = 24+ 2(68)/ 2(24) + 2(24) + 2(68)

 = 162/232

 = 0.69

Hardy Weinberg Equation

 p2 + 2pq + q2 = 1

 (0.31)2 + 2(0.69)(0.31)+ (0.69)2 = 1

0.0961 + 0.4278 + 0.4761 =1

The sum of the entries is p2 + 2pq + q2 = 1, as the genotype frequencies must sum to one. 

Hardy Weinberg Equation has been reported in article.

Comment # 2. Some data in the table1 is confusing. For Glucose, obese vs healthy is 94.5 ± 9.1 vs 94.5± 9.2, but p<0.0001, that won’t be possible. The Total cholesterol in control is 105.5 ± 13.6, but the Triglycerides is 104.5± 11.9, it was also impossible, because the Triglycerides is normally about 30% of the Total cholesterol, in this table, it is more than 90%. I would not report the data if I can’t confirm the accuracy.

Response: Thanks you, yes sure this was serious mistake correction has been made in the table While triglyceroids values has been also corrected by reconsidering units conversion and calculation from initial data sets. 

Comment # 3. According to the date of age, two groups was not age-matched, but the difference of age has not reached a significant level (p>0.12) only. Because the age may have a large influence on BMI, they should consider to be covariates in the correlation analysis or ANOVA analysis.

Response; yes sure age influences BMI of an individual therefore we recruited age, sex matched participants for this study as earlier we mentioned this in the result section but now we have also made it clear in material and methods section.

Comment # 4. If the authors choose the Dominant model in genotype analysis, it also should be used in the whole manuscript.

Response; Thank you for your valued input we have expanded the applied dominant model to the whole manuscript. 

Comment # 5. Two-way ANOVA used in table 1? What is the second independent variable besides obesity? Indeed, a two-way ANOVA with obesity and genotype of Asn107lle should be performed with the dependent variables including BMI, NPS level, and other serum indexes in table1 (if they are corrected). There is no need to compare the NPS level between obesity and control in different genotypes unless there has a significant interaction of obesity and genotype on the NPS level.

Response; We are grateful for your direction, that helped to analyses the data more effectively. Secondly we agree with you and have removed the comparison of NPS level between obese and controls in revised submission.

Comment # 6. The authors reported rs=0.884, p=0.0004 in the result (what is the meaning of rs?), but in the figure, they show r2=0.1797, p<0.0001? And it also seemed there did not have 116 plots on the figure, please confirmed and provided a more cleared one.

Response;.Yes, thank you again actually it is r2 values that was wrong presentation now has been fixed. 

Comment # 7. Molecular simulation can only provide a hypothesis that the SNP may influence the interaction of NPS and Asn107lle. But no real data supported the interaction was affected, and furthermore, this effect has an influence on obesity. Indeed, the authors should perform a moderation effect analysis using the genotype as a moderating variable or perform a mediation effect analysis using the NPS as a mediating variable, these analyses can check if the interaction of genotypes and NPS has an effect on obesity.

Response; Yes, we agree with your suggestion, Moderation effect analysis by A.Fhayes really is useful tool for such analyses. We have added section in results and Supplementary tableS2.

Comment # 8. Too many figures. Figure 1 does not need to be provided in the main text; Figure 2 and 3 need to be re-figured according to a new two-way ANOVA analysis; Figure5-7 can be put together into one figure.

Response. Yes right we reorganized the figures accordingly.

Comment # 9. The simulation is just a prediction; I would not write three long paragraphs to report these results, even longer than the results from real data.

 Response. Yes we have summarized the simulation data in revised submission. 

Comment # 10. The RESULTS may be better presented with separate paragraphs accompanied by sub-titles.

Response. Yes we have organized our results accordingly. Thank you

---

## [Decision Letter · Decision Letter 2]

1 Sep 2020

PONE-D-19-35919R2

Neuropeptide S receptor gene Asn107 polymorphism in obese male individuals in Pakistan

PLOS ONE

Dear Dr. Bokhari,

Thank you for submitting your manuscript to PLOS ONE. After careful consideration, we feel that it has merit but does not fully meet PLOS ONE’s publication criteria as it currently stands. Therefore, we invite you to submit a revised version of the manuscript that addresses the points raised during the review process.

We look forward to receiving your revised manuscript.

Kind regards,

Zezhi Li, Ph.D., M.D.

Academic Editor

PLOS ONE

Reviewers' comments:

Reviewer's Responses to Questions

**Comments to the Author**

1. If the authors have adequately addressed your comments raised in a previous round of review and you feel that this manuscript is now acceptable for publication, you may indicate that here to bypass the “Comments to the Author” section, enter your conflict of interest statement in the “Confidential to Editor” section, and submit your "Accept" recommendation.

Reviewer #2: (No Response)

2. Is the manuscript technically sound, and do the data support the conclusions?

Reviewer #2: Partly

3. Has the statistical analysis been performed appropriately and rigorously? 

Reviewer #2: No

4. Have the authors made all data underlying the findings in their manuscript fully available?

Reviewer #2: Yes

5. Is the manuscript presented in an intelligible fashion and written in standard English?

Reviewer #2: Yes

6. Review Comments to the Author

Reviewer #2: Comment1#:

The author does not understand the meaning of Hardy Weinberg's equation, and the formula replied by the author can only prove that our basic laws of mathematics have not yet failed. Deviation from HWE means that there is a significant difference between the genetic distribution of the sample and the theoretical distribution which calculated based on the least allele frequency; such difference is generally analyzed by using chi-square test, and the P value of this test needs to be reported. For more details about HWE, authors can refer to genetic textbooks or Wikipedia. With regard to the calculation of HWE, authors can use software such as SPSS or some online calculators on the Internet.

Comment2#:

According to your previous data, this is not an age-matched case-control study. Age-match means that the age is the same, not similar, between case and control. Therefore, I suggest to remove the statement of age-matched. Furthermore, the data of age should not be deleted in table1. The author also did not use age as a covariable as I suggested previously. Is it because the correlation no longer significance after age controlled? Even so, please report the results with covariates and discuss the results. Because it directly affects the conclusion.

Comment3#

I suggested using obesity and genotype as independent variables, and BMI, NPS and blood indicators as dependent variables to perform a two-way ANOVA, but the authors did not do that. The influence of genotype on these indexes (BMI, NPS, and blood indicators) is need to be reported, which can be shown by two-way ANOVA, and it can also show whether there is interaction between obesity and genotypes on different indicators.

Comment4#

The data reported about moderation effect analysis is confusing. There are three results should be reported, the effects of BMI, the genotype and their interactions on NPS respectively. The interaction seemed not be reported. The significant moderation of genotype means that the correlations between BMI and NPS are significant difference between the two genotypes, so please report the correlation coefficient between BMI and NPS in the two genotypes respectively, and make a figure for it.

Comment5#

The peak map of sequencing does not need to be provided in the main body of the manuscript, because it does not show the data, but use to prove the reliability of the genotyping.

7. PLOS authors have the option to publish the peer review history of their article (what does this mean?). If published, this will include your full peer review and any attached files.

Reviewer #2: No

---

## [Author Response · Author response to Decision Letter 2]

28 Oct 2020

Response Letter addressing “Revision”

Dear Editor PLOS ONE,

With Reference ;

PONE-D-19-35919R2

Neuropeptide S receptor gene Asn107 polymorphism in obese male individuals in Pakistan

The Manuscript Revision comments & Response here for your consideration. The authors appreciate the time the reviewers have invested in reviewing our manuscript carefully and for their valuable input which has significantly enhanced the impact of this manuscript. Following are the responses to reviewer comments and subsequent changes in the revised manuscript.

The specific comments and their responses are mentioned below.

Reviewer #2: Comment1#:

The author does not understand the meaning of Hardy Weinberg's equation, and the formula replied by the author can only prove that our basic laws of mathematics have not yet failed. Deviation from HWE means that there is a significant difference between the genetic distribution of the sample and the theoretical distribution which calculated based on the least allele frequency; such difference is generally analyzed by using chi-square test, and the P value of this test needs to be reported. For more details about HWE, authors can refer to genetic textbooks or Wikipedia. With regard to the calculation of HWE, authors can use software such as SPSS or some online calculators on the Internet.

Response:

Methods:

We thank the reviewer for this extremely helpful suggestion. We have implemented this in the new manuscript as follows:

The allele frequency was obtained by counting the number of alleles in the total population and dividing by twice the number of study subjects. For example, AA has two A alleles, AT has one A allele and one T allele and TT has two TT alleles. The expected genotype probability was calculated using the following table based upon the Hardy-Weinberg equation with p being the probability of the A allele and q being the probability of the T allele. 

 A (p) T (q)

A (p) AA(p2) AT(pq)

T (q) AT(qp) TT (q2)

The genotype frequencies were then multiplied by the number of study subjects to get the expected genotype frequencies. The chi-squared statistic was derived using standard methods using Excel.

Results:

Using the allele frequency calculate from the collected data, the expected distribution of genotypes was calculated using the Hardy-Weinberg equation. These genotype frequencies were used as the expected values in a chi-squared test to see if the observed genotype frequency is significantly different from the expected frequency. The allele frequency in the observed population is 0.69 for A and 0.31 for T (160/232=0.69 for A and 72/232=0.31 for T). The frequency for the genotypes were calculated and are shown below. 

 Expected-genotype 

 Frequency

 A T

 A 0.47562426 0.21403092

 T 0.21403092 0.09631391

The resulting chi-squared tables for observed and expected for the null hypothesis that the genotype frequencies are the same are 

 observed expected 

 obese control obese control 

AA 28 40 68 AA 27.11058 28.0618312 55.1724138

TT 14 10 24 TT 5.489893 5.68252081 11.1724138

AT 15 9 24 AT 24.39952 25.255648 49.6551724

 57 59 116 57 59 116

The chi-squared statistic with 2 degrees of freedom is 35.66 which is greater than 10.597 (p=0.005). Therefore, we reject the null hypothesis that the observed and expected genotype frequencies ae the same. 

Comment2#:

According to your previous data, this is not an age-matched case-control study. Age-match means that the age is the same, not similar, between case and control. Therefore, I suggest to remove the statement of age-matched. Furthermore, the data of age should not be deleted in table1. The author also did not use age as a covariable as I suggested previously. Is it because the correlation no longer significance after age controlled? Even so, please report the results with covariates and discuss the results. Because it directly affects the conclusion.

Response: 

We have corrected the manuscript as suggested. The age-matched statement has been replaced and age data has been restored in table 1. Furthermore Age has been considered as variable in two way ANOVA analysis. 

Comment3#

I suggested using obesity and genotype as independent variables, and BMI, NPS and blood indicators as dependent variables to perform a two-way ANOVA, but the authors did not do that. The influence of genotype on these indexes (BMI, NPS, and blood indicators) is need to be reported, which can be shown by two-way ANOVA, and it can also show whether there is interaction between obesity and genotypes on different indicators.

Response: 

Methods: 

We appreciate this thoughtful suggestion. We have implemented the two-way ANOVA as follows:

 A two-way Analysis Of Variance (ANOVA) was performed to test if the means of the measured dependent variables NPS, triglycerides, blood sugar, BMI, HDL, LDL, age, and cholesterol were different for different values of the independent variables, obesity and genotype. The Tukey’s Honestly-Significant-Difference (TukeyHSD) test was performed as a post-hoc procedure to see which groups are different from one another. Statistics were done using R 3.6.31, RStudio 1.3.1093 2, the car3, the rstatix4, and the emmeans5 packages. Figures were produced using the ggplot26 and the ggpubr7 packages. The full reproducible code is available in Supplementary Materials.

Results:

We applied two way ANOVA modelto compare the means of independent variables obesity and genotype against the dependent variables Age, NPS and lipid profile parameters both in obese and control groups. 

 In this study we found the means for NPS and triglycerides were significantly (p=0.0001) different for different genotypes in obese and control group. Furthermore, means for Cholesterol, BMI, HDL, blood sugar and LDL are significantly different in obese and control group (p<0.01). Interestingly only in the case of triglycerides there is an interaction between the independent variables obesity and genotype.

Table 3 also shows the Tukey HSD p-values for the two-way analysis. These values indicate that we reject the hypothesis that for NPS AT and AA have equal means and that TT and AA have equal mean. We found a statistically significant difference in average NPS by both SNP (f(2)= 26.0613, p <5.435e-10 ) and by obesity (f(1)= 186.6127, p<2.2e-16), though the interaction between these terms was not significant. A Tukey post-hoc test revealed significant pairwise differences between genotype “AA” and genotype “AT” (diff= 47.930945), between “AA” and genotype TT” (diff= 47.930945). No significant difference was found between “AT” and genotype TT” (diff = -2.531281). Moreover, We found a statistically significant difference in average Triglycerides by both SNP (f(2)= 4.7670, p = 0.01034) and by obesity (f(1)= 399.2172, p<2.2e-16); however the interaction between these terms was not significant. NPS and Triglycerides are the only dependent variables that show a significant difference in both SNP and obesity.

Comment4#

The data reported about moderation effect analysis is confusing. There are three results should be reported, the effects of BMI, the genotype and their interactions on NPS respectively. The interaction seemed not be reported. The significant moderation of genotype means that the correlations between BMI and NPS are significant difference between the two genotypes, so please report the correlation coefficient between BMI and NPS in the two genotypes respectively, and make a figure for it.

Response:

Method 

The moderation effect analysis was performed to analyze the effect of obesity as independent variable directly on NPS concentration considering genotype as moderator using PROCESS Procedure for SPSS Version 3.5 by Andrew F. Hayes.

PROCESS Procedure for SPSS Version 3.5 ***************** 

 Written by Andrew F. Hayes, Ph.D. www.afhayes.com

 Documentation available in Hayes (2018). www.guilford.com/p/hayes3 

Model : 1 

 Y : NPS 

 X : Obesity 

 W : Genotype 

Sample 

Size: 116 

OUTCOME VARIABLE: 

 NPS 

Model Summary 

 R R-sq MSE F(HC4) df1 df2 p 

 .691 .478 1962.044 31.460 3.000 112.000 .000 

Model 

 coeff se(HC4) t p LLCI ULCI 

constant 121.352 15.050 8.063 .000 91.532 151.171 

Obesity -10.962 2.565 -4.273 .000 -16.045 -5.879 

Genotype 17.447 8.617 2.025 .045 .373 34.521 

Int_1 2.873 1.432 2.006 .047 .035 5.711 

Product terms key: 

 Int_1 : Obesity x Genotype 

Test(s) of highest order unconditional interaction(s): 

 R2-chng F(HC4) df1 df2 p 

X*W .020 4.022 1.000 112.000 .047 

---------- 

 Focal predict: Obesity (X) 

 Mod var: Genotype (W) 

Conditional effects of the focal predictor at values of the moderator(s): 

 Genotype Effect se(HC4) t p LLCI ULCI 

 AA -8.089 1.229 -6.583 .000 -10.524 -5.654 

 AT/TT -5.216 .736 -7.085 .000 -6.675 -3.757 

Data for visualizing the conditional effect of the focal predictor: 

Paste text below into a SPSS syntax window and execute to produce plot. 

DATA LIST FREE/ 

 Obesity Genotype NPS . 

BEGIN DATA. 

 -6.275 1.000 189.558 

 .000 1.000 138.798 

 6.275 1.000 88.039 

 -6.275 2.000 188.977 

 .000 2.000 156.245 

 6.275 2.000 123.513 

END DATA. 

GRAPH/SCATTERPLOT= 

 Obesity WITH NPS BY Genotype .

Result

NPS concentrations were significantly (p=0.000) moderated by NPSR1 gene Asn107Ile polymorphism with increasing obesity (Fig 4). Furthermore moderation effect analyses revealed a more negative effect of obesity on NPS concentration with AT+TT vs AA genotype as shown in (Table S2). 

ANALYSIS NOTES AND ERRORS ************************ 

Level of confidence for all confidence intervals in output: 

95 

NOTE: A heteroscedasticity consistent standard error and covariance matrix estimator was used. 

NOTE: The following variables were mean centered prior to analysis: 

 Obesity 

------ END MATRIX ----- 

Comment5#

The peak map of sequencing does not need to be provided in the main body of the manuscript, because it does not show the data, but use to prove the reliability of the genotyping.

Response.

Yes Agreed, the Figure have been removed from the main body of text. Thanks 

References

1. R Core Team (2020). R: A language and environment for statistical computing. R Foundation for Statistical Computing, Vienna, Austria. URL https://www.R-project.org/.

2. RStudio Team (2020). RStudio: Integrated Development for R. RStudio, PBC, Boston, MA URL http://www.rstudio.com/.

3. John Fox and Sanford Weisberg (2019). An {R} Companion to Applied Regression, Third Edition. Thousand Oaks CA: Sage. URL: https://socialsciences.mcmaster.ca/jfox/ Books/Companion/.

4. Alboukadel Kassambara (2020). rstatix: Pipe-Friendly Framework for Basic Statistical Tests. R package version 0.6.0. https://CRAN.R-project.org/package=rstatix.

5. Russell Lenth (2020). emmeans: Estimated Marginal Means, aka Least-Squares Means. R package version 1.5.1. https://CRAN.R-project.org/package=emmeans.

6. H. Wickham. ggplot2: Elegant Graphics for Data Analysis. Springer-Verlag New York, 2016.

7. Alboukadel Kassambara (2020). ggpubr: “ggplot2” Based Publication Ready Plots. R package version 0.4.0. https://CRAN.R-project.org/package=ggpubr.

---

## [Decision Letter · Decision Letter 3]

18 Nov 2020

Neuropeptide S receptor gene Asn107 polymorphism in obese male individuals in Pakistan

PONE-D-19-35919R3

Dear Dr. Bokhari,

We’re pleased to inform you that your manuscript has been judged scientifically suitable for publication and will be formally accepted for publication once it meets all outstanding technical requirements.

Kind regards,

Zezhi Li, Ph.D., M.D.

Academic Editor

PLOS ONE

Additional Editor Comments (optional):

Reviewers' comments:

Reviewer's Responses to Questions

**Comments to the Author**

1. If the authors have adequately addressed your comments raised in a previous round of review and you feel that this manuscript is now acceptable for publication, you may indicate that here to bypass the “Comments to the Author” section, enter your conflict of interest statement in the “Confidential to Editor” section, and submit your "Accept" recommendation.

Reviewer #2: All comments have been addressed

2. Is the manuscript technically sound, and do the data support the conclusions?

Reviewer #2: Partly

3. Has the statistical analysis been performed appropriately and rigorously? 

Reviewer #2: I Don't Know

4. Have the authors made all data underlying the findings in their manuscript fully available?

Reviewer #2: Yes

5. Is the manuscript presented in an intelligible fashion and written in standard English?

Reviewer #2: (No Response)

6. Review Comments to the Author

Reviewer #2: (No Response)

7. PLOS authors have the option to publish the peer review history of their article (what does this mean?). If published, this will include your full peer review and any attached files.

Reviewer #2: No

---

## [Editor Report · Acceptance letter]

26 Nov 2020

PONE-D-19-35919R3 

Neuropeptide S receptor gene Asn107 polymorphism in obese male individuals in Pakistan 

Dear Dr. Bokhari:

I'm pleased to inform you that your manuscript has been deemed suitable for publication in PLOS ONE. Congratulations! Your manuscript is now with our production department. 

Kind regards, 

on behalf of

Dr. Zezhi Li 

Academic Editor

PLOS ONE